# Plasma-Driven Photocatalysis Based on Gold Nanoporous Arrays

**DOI:** 10.3390/nano11102710

**Published:** 2021-10-14

**Authors:** Lisheng Zhang, Yiyuan Zhang, Xueyan Wang, Duan Zhang

**Affiliations:** The Beijing Key Laboratory for Nano-Photonics and Nano-Structure, Department of Physics, Capital Normal University, Beijing 100048, China; 2190602008@cnu.edu.cn (Y.Z.); xueyanwang@cnu.edu.cn (X.W.); duanzhang@cnu.edu.cn (D.Z.)

**Keywords:** gold nanoporous arrays, surface plasmon, Raman, photocatalysis

## Abstract

Various effects caused by surface plasmons including enhanced electromagnetic field, local heating, and excited electrons/holes can not only redistribute the electromagnetic field in the time domain and space but also redistribute the excited carriers and drive chemical reactions. In this study, firstly, an Au nanoporous array photocatalyst with the arrayed gauge was prepared by means of the anodic alumina template. Then, the formation of 4,4′-dimercaptoazobenzene (DMAB) by the surface plasmon-driven photocatalysis under 633 nm laser irradiation was investigated by means of Raman spectroscopy using aminothiophenol (PATP) as a probe molecule on gold nananoporous arrays. In addition, sodium borohydride was introduced in situ to realize the reverse photocatalytic reaction driven by the surface plasma. With the help of FDTD software, the plasma distribution characteristics on the surface of Au nanoporous arrays were simulated and analyzed. Through this practical method, it is expected to draw specific graphics, letters, and Chinese characters on the micro/nano scale, and realize the functions of graphics drawing, information encryption, reading, and erasing on the micro/nano scale.

## 1. Introduction

Surface enhanced Raman spectroscopy (SERS) is a common spectral technique in which the inelastic light scattering of molecules is greatly enhanced. SERS can enhance the weak Raman signal by 10^5^–10^8^, providing detailed structural information and binding properties of molecules on the surface. Since the discovery of SERS, the interest in SERS has been steadily growing and many other spectral techniques have been introduced. These techniques utilize the enhanced local field generated by surface plasmon excitation to apply to optical phenomena such as fluorescence or nonlinearity [1,2,3,4,5,6]. In addition, the coupling of SERS with the tip of atomic force microscope (AFM) or scanning tunneling microscope (STM) makes tip-enhanced Raman scattering (TERS) a powerful imaging tool. For analytical applications, SERS provides abundant vibration spectrum information, which makes it different from other technologies and makes it widely used in various fields, including electrochemistry, catalysis, biology, medicine, art protection, material science, etc. [7,8,9,10,11,12].

The enhancement mechanism of SERS has been known as one of the prevalent topics in this field. The Raman signal enhancement of the probe molecules adsorbed on the metal nanostructures comes from two factors including the enhancement of the electromagnetic field on the metal surface and increase of charge transfer between the metal center and molecules. These two factors are attributed to the electromagnetic enhancement (EM) and chemical enhancement (CE) mechanisms, respectively [13,14,15]. The EM enhancement is defined as the collective oscillation of conduction electrons on the surface of metal nanostructures, which is called local surface plasmon resonance (LSPR). The plasmons in the noble metal nanostructures can be significantly amplified relative to the incident light intensity. Although SERS can be obtained from the electric field enhancement at a single nanoparticle (NP), it is advantageous to design more fine nanostructures for SERS. By placing molecules in the nanoscale gap between the two metal NPs (the so-called hot spot), the Raman signal of the probe molecule will be further enhanced to obtain a higher SERS enhancement factor.

Plasma-driven surface catalytic oxidation-reduction reactions have attracted wide attention since they were first reported. By consideration of its working principles to its wide applications in physics, chemistry, environmental science, material science, and other fields, several reaction mechanisms have been proposed for plasma-driven surface catalytic oxidation-reduction reactions. The hot electrons produced by plasma decay play a key role and not only provide the electrons needed for the reduction reaction but also induce a large amount of kinetic energy to overcome the potential energy barrier.

As an important tool to study plasma driven photocatalytic reactions, SERS can not only detect the chemical information of probe molecules, but also provide suitable catalytic substrates for surface catalytic reactions [16,17,18,19]. Wu et al. [15] reported that PATP adsorbed on the surface of metal nanoparticles can be catalyzed to form 4,4′-dimercaptoazobenzene (DMAB) by LSPR [20]. Our previous work has also proven that PATP can be oxidized to DMAB in the micro-region by using TERS [21]. In the present study, the real-time monitoring of the formation of DMAB by the photocatalytic reaction of PATP, as a probe molecule driven by local surface plasma, was investigated based on the regular arrangement of gold nanoporous array catalytic substrate and using the advantage of surface enhanced Raman spectroscopy. After that, sodium borohydride was introduced in an in situ manner. Under the same experimental conditions, the product DMAB was driven by plasma to produce PATP again.

## 2. Materials and Methods

Firstly, according to the two-step anodization method mentioned in the literature, the hexagonal anodic aluminum oxide template (AAO) was fabricated [22]. Then, a layer of gold with moderate thickness was deposited on the surface of the AAO template by vacuum thermal evaporation technology and a gold nanoporous array substrate with a regular hexagonal arrangement was obtained. The concentration of PATP was 10^−3^ mol/L^−1^ in absolute ethanol. Then, the prepared gold nanoporous array substrate was placed in a PATP solution. After standing for 1 h, the substrate was taken out and naturally dried for subsequent collection of surface plasmon-driven photocatalytic Raman spectra. The Au nanoarray substrate with PATP probe molecules was placed in the Raman spectrometer (HORIBA-HR800, Tokyo, Japan). The Raman spectra were collected with 633 nm excitation, 50 × objective, and 1 mW laser power at the exit of the objective. In order to avoid the catalysis of PATP under natural light, the whole experimental process needed to be carried out in a dark room.

## 3. Results

In this work, we used PATP as a probe to study the surface plasma-driven photocatalytic reaction based on the gold nanoporous arrays. Figure 1a shows the structure and Raman spectrum of PATP probe molecule. The black line is the Raman spectrum simulated by Gaussian 09 software and the red line is the Raman spectrum collected by PATP solid powder. It can be seen from this figure that the Raman peak at 1084 cm^−1^ is a significant characteristic peak of PATP molecule. Figure 1b shows the structure of DMAB molecule and the Raman spectrum characteristic peaks obtained by the experimental and theoretical calculations. By comparing Figure 1a,b, it can be seen that the Raman peaks of PATP at 1084 and 1590 cm^−1^ and those of DAMB at 1142, 1388, and 1440 cm^−1^ were not repeated. Therefore, in the next experiment, the Raman characteristic peaks at 1084 cm^−1^ can be used to characterize the existence of PATP, and the Raman peaks at 1142, 1388, and 1440 cm^−1^ can be used to indicate the existence of DMAB. Figure 1c is a schematic diagram of the photocatalytic reaction process driven by the surface plasma of the gold nanoporous array substrate. Firstly, the gold nanoarray substrate was immersed in the PATP solution and the PATP molecules were uniformly adsorbed on the gold surface; then, the sulfur atoms in the PATP molecules formed gold sulfur bonds with the gold atoms. When the surface of the substrate was irradiated by 633 nm laser, the region around the nanoporous produced a very strong local surface plasmon-enhanced hot spot region due to the regular hexagonal arrangement of Au nanoporous. Compared with the excitation light itself, the intensity of the excitation photoelectric magnetic field in the hot spot was greatly enhanced, and the nitrogen hydrogen bond of PATP probe molecule in the hot spot was broken to form a nitrogen double bond under the dual action of surface plasmon and excitation light. Thus, the photocatalytic reaction took place to form a new structure of DMAB molecule. After that, the product DMAB produced in the early stage underwent a reverse photocatalytic reaction by dropping 10^−2^ mol/L^−1^ anhydrous ethanol solution of sodium borohydride in situ, under the action of surface plasmon, stimulated luminescence, and sodium borohydride to form a PATP molecule.

Figure 2 is the morphology analysis of the AAO template and gold nanoporous array. Figure 2a,b are AFM pictures of the AAO template. It can be seen from these pictures that the surface of the template is very flat and nanoporous, and the nanopores are arranged in a hexagonal shape with a uniform pore size. Figure 2c is the SEM image of gold nanoporous array prepared by the vacuum evaporation coating technology based on the AAO template. It can be seen that the distribution of gold nanoporous array is regular, which is consistent with that of AAO. The diameter of the prepared gold nanoporous is about 75 nm, and the pore size distribution is uniform. It can be seen from the comprehensive analysis in Figure 2 that the gold nanoporous array with the regular arrangement has a flat surface, uniform size, and invariable hot spot distribution, which is an ideal catalytic substrate for the study of surface plasma-driven photocatalytic reactions.

However, in fact, the PATP probe molecules located on the surface of gold nanoporous array substrate, as shown in Figure 1c, may be effective in the photocatalysis and reverse photocatalysis under the action of 633 nm excitation and sodium borohydride, respectively. It can be verified by the real-time collection of Raman spectra and identification of characteristic peaks in the Raman spectra. Figure 3 shows the Raman spectra of PATP probe molecules based on the gold nanoporous array catalyst substrate in the two stages of photocatalytic reaction under the action of 633 nm laser and reverse catalytic reaction due to the in situ introduction of sodium borohydride. Figure 3a displays the Raman spectra collected in real time at the first stage. The five spectral lines in this figure from bottom to top are the Raman spectra collected every 2 s, when the 633 nm laser was focused on the surface of the substrate in a dark room. These are the spectral lines when the laser continuously acted on the photocatalytic system for 2, 4, 6, 8, and 10 s. According to the previous analysis, the Raman peaks at 1084 cm^−1^ can be attributed to the characteristic peaks of PATP and the Raman peaks at 1142, 1388, and 1440 cm^−1^ can be attributed to the characteristic peaks of DMAB. From the bottom to top, the spectral lines in Figure 3a show that the Raman characteristic peak of PATP molecule at 1084 cm^−1^ always exists, which can prove that PATP always exists in the first stage of the photocatalytic process. This means that the PATP molecule is excessive in the whole process. The Raman peaks at 1142, 1388, and 1440 cm^−1^ appeared at the laser irradiation for 2 s and their intensities gradually increased with time. The characteristic peak of DMAB located at 1142 cm^−1^ in Figure 3a is fitted with the peak intensity of area, as shown in Figure 3b. This figure shows that the intensity of the characteristic Raman peak confirms the presence and relative content of DMAB at 1142 cm^−1^, as increased gradually with time. It can be seen that PATP molecules on the gold nano-array catalytic substrate have a very rapid photocatalytic reaction to generate DMAB molecules under the action of a 633 nm laser. With the extension of time and increasing concentration of DMAB, the PATP molecules were continuously converted into DMAB molecules.

Figure 3c shows the Raman spectra collected under the same experimental conditions by adding sodium borohydride solution in situ after the first photocatalytic reaction. The 1084 cm^−1^ Raman characteristic peak of PATP molecule was always present from bottom to top, which proves the existence of PATP in the entire second stage. The intensity of Raman peaks at 1142, 1388, and 1440 cm^−1^ decreased rapidly and disappeared with time under the action of sodium borohydride and laser. At the same time, there are no new Raman peaks, which means that there are no new molecules except for PATP and DMAB. It can be seen that the firstly generated photocatalytic product DMAB reverses the photocatalytic reaction in the presence of sodium borohydride and laser to generate the PATP molecules. Through the analysis of Figure 2, it can be concluded that the PATP probe molecules located on the surface of gold nanoporous array substrate had a rapid photocatalytic reaction under the action of 633 nm excitation light and hot spots on the gold nanoporous array; therefore, the product DMAB molecules can be generated continuously and rapidly. Subsequently, the reverse photocatalytic reaction of DMAB molecules on the surface of Au nanoporous arrays could be catalyzed by sodium borohydride, 633 nm excitation light, and hot spots on the Au nanoporous arrays, therefore, the PATP molecules were generated again.

The FDTD software was used to simulate the surface plasmon distribution on the surface of gold nanoarray (as shown in Figure 4). Figure 4a is a theoretical model based on the geometric dimensions of AFM and SEM images of the prepared gold nanohole array structure substrate and the polarization direction of surface plasmon-stimulated luminescence. Figure 4b shows calculation results of the surface plasmon intensity distribution characteristics of gold nanohole array by the FDTD software. It can be deduced that strong local surface plasmon enhancement hot spots are generated in the area around the edge of gold nanohole array under the effect of excitation light. These hot spots are regularly arranged according to the distribution of nanoporous, and are dependent on the polarization direction of the excitation light. Figure 4c shows the relationship between the electromagnetic field intensity and ‘x’ on the straight line at ‘y’ = 0 in the calculation results. It can be concluded that the electromagnetic field intensity on the surface between the gold nanoholes is very low, and the area near the inner edge of the nanoholes is very strong. At the same time, the intensity state depends on the arrangement of the nanoholes and presents a periodic distribution. There are 6 subfigures numbered from 1 to 6 in Figure 4d. They show the changes of the spatial distribution characteristics of the surface plasmon with time after the surface of the gold nanohole array was subjected to an induced light surface wave. It can be seen that the localized surface plasmon oscillated along the inner edge of the Au nanohole and gradually decreased until it disappeared.

Figure 4e is a schematic diagram of the photocatalytic mechanism driven by the surface plasma of the gold nanoporous array substrate. In the figure, ‘E_f_’ is the Fermi level of gold; ‘E_1_′ is the energy of a single photon; ‘LUMO’ and ‘HOMO’ are the ‘Lowest Unoccupied Molecular Orbital’ and ’Highest Occupied Molecular Orbital’ of PATP, respectively; and ‘h^+^’ is the energy distribution of hot electrons excited by 633 nm laser. This diagram denotes that the energy required for the photocatalytic reaction of PATP mainly comes from three aspects. The first is that PATP molecules are in contact with gold nanoporous arrays, and its electrons are affected by the gold Fermi energy level, which improves their energy. In the second part, the energy directly comes from the absorption of 633 nm excited photons. The third part of the energy source is due to the absorption of the energy of hot electrons and holes produced by the decay of surface plasmon emitted by 633 nm laser.

## 4. Conclusions

In summary, an Au nanoporous array catalytic substrate with regular arrangement was prepared based on AAO template. When the specific wavelength of the excitation light acted on the substrate, a large number of regularly arranged local surface plasmon enhancement regions were generated on its surface. With the advantage of SERS, we can monitor the photocatalytic reaction of PATP to DMAB in a real time. After that, the in situ introduction of sodium borohydride simplified the reaction of DMAB with the plasma to form a pair of PATP molecules under the same experimental conditions. This research will provide a new route for the drawing and erasing of molecular graphics on micro and nano scales, as well as significant applications in information encryption, reading, and erasing processes.

## Figures and Tables

**Figure 1 nanomaterials-11-02710-f001:**
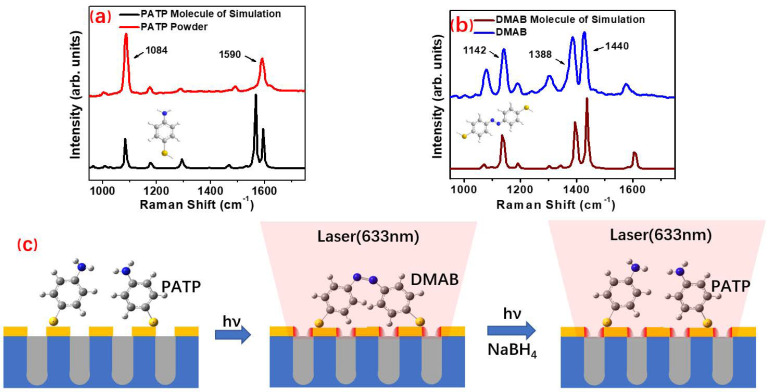
The formation process of DMAB from PATP surface plasma driven photocatalytic reaction: (**a**) PATP Raman characteristic peak; (**b**) DMAB Raman characteristic peak; (**c**) the schematic diagram of the plasma driven photocatalytic reaction.

**Figure 2 nanomaterials-11-02710-f002:**
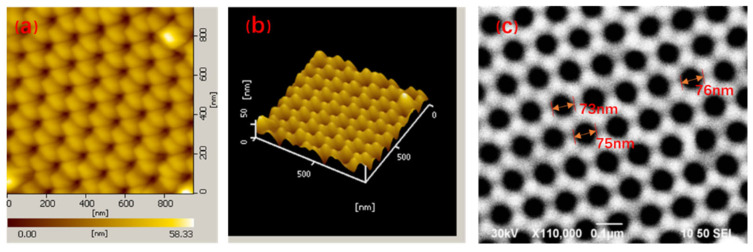
Surface topography of gold nanoporous arrays substrate: (**a**) 2D AFM image of gold nanoporous arrays; (**b**) three-dimensional AFM image of nanoporous arrays; (**c**) SEM image of nanoporous arrays.

**Figure 3 nanomaterials-11-02710-f003:**
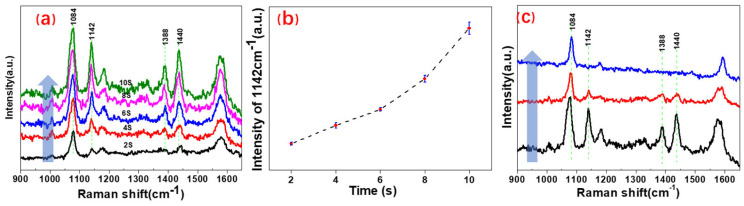
Raman spectra of plasma-driven photocatalysis based on gold nanohole array substrate: (**a**) Raman spectrum of PATP photocatalysis generating DMAB; (**b**) the change trend of Raman peak intensity at 1142 cm^−1^ with time; (**c**) Raman spectrum of DMAB reverse reaction generating PATP.

**Figure 4 nanomaterials-11-02710-f004:**
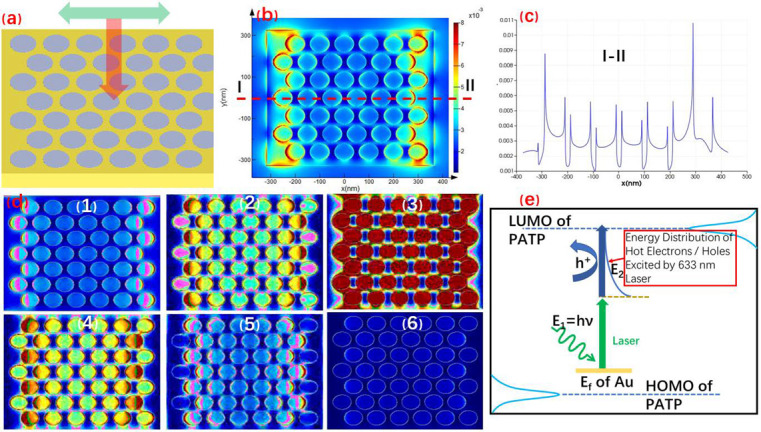
The mechanism of gold nanohole array substrate plasma-driven photocatalysis is as follows: (**a**) theoretical model of gold nanohole array and polarization direction of luminescence simulated by the surface plasmon; (**b**) calculation results of intensity distribution characteristics of the surface plasmon of gold nanohole array; (**c**) analysis of intensity distribution simulated by the surface plasmon of gold nanohole array; (**d**) surface plasmon distribution of gold nanohole array; (**e**) the mechanism of Au nanohole array substrate plasma-driven photocatalysis.

## Data Availability

The data presented in this study are available on request from the corresponding author.

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
