# Peer review of "Plasma-Driven Photocatalysis Based on Gold Nanoporous Arrays"

_nanomaterials, 2021, doi:10.3390/nano11102710_

Round 1

Reviewer 1 Report

The manuscript treats the plasma driven photocatalysis based on gold nanopore arrays is rather well structured. The english style is average except at some parts where, the phrasing should be improved. Detailed comments are the following:

-L 54: the sentence can be rephrased as follows: The hot electrons produced by plasma decay play...

- In the abstract PATP is said to be the abbreviation of aminothiophenol while in l 59 the correct world is para-aminothiophenol .

-L 57-58. To rephrase.

-L 59. Add the reference just after Wu et al.

-L 80. You can say "In order to avoid the catalysis of PATP...."

-L 107. "...was undergo..." to improve since its meaning is not clear.

-L 118-119. To avoid repetition of "arranged".

-L 145-146. represent --> show

-L 149-150. "were appeared" and "were gradually increased". These mean that someone on some thing act to make appear and to increase the subject. It is better to remove "were" in  both cases.

-L 167. "was decreased". Same remark as the previous one.

-Fig 4. in (d) there are 6 subfigures numbered from 1 to 6. It is better to explain these subfigures in the text. In figure 4.e the words LUMO and HOMO are not explained as well as h+ (small H). Same thing for E1 and E2.In the caption of fig 4, "simulated" is repeated.

Reviewer 2 Report

In this paper the authors report on the use of a plasmonic platform to drive the photocatalytic reaction between PATP and DMAB.

they used SERS to monitor the process.

the manuscript is well written and to me rather interesting.

I do not see any particular criticism.

my only recommendation is to modify the terminology. This is not a plasmonic nanopore platform (a nanopore is an aperture that connect two separated compartments), but it can be defined as a "nanoporous"plasmonic platform. Regarding to this I recommend the authors to mention a recent paper on this topic where plasmonic photocatalysis is also discussed 

ACS Nano 2021, 15, 4, 6038–6060
